# Identification of Human Global, Tissue and Within-Tissue Cell-Specific Stably Expressed Genes at Single-Cell Resolution

**DOI:** 10.3390/ijms231810214

**Published:** 2022-09-06

**Authors:** Lingyu Qiu, Chen Liang, Yidong Zheng, Huayu Kang, Aiyue Chen, Chunlin Chen, Xinlong Wang, Jielin Yang, Qiongfang Fang, Xinjie Hui, Yueming Hu, Zewei Chen, Ou Sha, Wei-Guo Zhu, Yejun Wang

**Affiliations:** 1Guangdong Key Laboratory of Genome Instability and Human Disease Prevention, Department of Biochemistry and Molecular Biology, Shenzhen University School of Medicine, Shenzhen 518060, China; 2Youth Innovation Team of Medical Bioinformatics, Shenzhen University Health Science Center, Shenzhen 518060, China; 3School of Stomatology, Shenzhen University Health Science Center, Shenzhen 518060, China; 4International Cancer Center, Shenzhen University School of Medicine, Shenzhen 518060, China; 5Shenzhen Bay Laboratory, Shenzhen University School of Medicine, Shenzhen 518060, China

**Keywords:** Stably Expressed Gene (SEG), Housekeeping Gene (HKG), single cell RNA-seq (scRNA-seq), cell decomposition

## Abstract

Stably Expressed Genes (SEGs) are a set of genes with invariant expression. Identification of SEGs, especially among both healthy and diseased tissues, is of clinical relevance to enable more accurate data integration, gene expression comparison and biomarker detection. However, it remains unclear how many global SEGs there are, whether there are development-, tissue- or cell-specific SEGs, and whether diseases can influence their expression. In this research, we systematically investigate human SEGs at single-cell level and observe their development-, tissue- and cell-specificity, and expression stability under various diseased states. A hierarchical strategy is proposed to identify a list of 408 spatial-temporal SEGs. Development-specific SEGs are also identified, with adult tissue-specific SEGs enriched with the function of immune processes and fetal tissue-specific SEGs enriched in RNA splicing activities. Cells of the same type within different tissues tend to show similar SEG composition profiles. Diseases or stresses do not show influence on the expression stableness of SEGs in various tissues. In addition to serving as markers and internal references for data normalization and integration, we examine another possible application of SEGs, i.e., being applied for cell decomposition. The deconvolution model could accurately predict the fractions of major immune cells in multiple independent testing datasets of peripheral blood samples. The study provides a reliable list of human SEGs at the single-cell level, facilitates the understanding on the property of SEGs, and extends their possible applications.

## 1. Introduction

Housekeeping genes (HKGs) are genes expressed universally within different tissues, under different conditions and at various development stages [1]. Typically, HKGs are evolutionarily conserved and abundantly expressed, participating in various fundamental biological processes [2]. Some research groups also proposed ‘stably expressed genes’ (SEGs) recently, which describe a subset of genes similar to HKGs, but which emphasize invariant expression quantitatively [1,3].

It is of great significance to explore HKGs or SEGs, their properties, evolution, functional relevance and potential application. HKGs identified by different studies, albeit heterogeneously for gene composition and organisms, disclosed unique gene structure and evolutionary characteristics; for instance, shorter introns and exons [4], lower conservation of the promoter sequence [5], less potential for nucleosome formation in the 5’region [6], and enrichment of protein products in specific domain families [7]. In addition to biological interest, HKGs and especially SEGs have more practical applications. Since the concept of HKGs was proposed, these genes have been used as internal control for individual gene quantification or large-scale gene-expression profiling experiments [8,9]. A list of HKGs have been applied as quantification references in biological and medicine related studies widely, e.g., glycated hyde-3-phosphorylated hydrogenase (GAPDH), β-actin, β-tubulin, ubiquitin C, β2-microglobulin (B2M), 18S rRNA, etc. [10,11,12,13]. Due to systematic errors such as PCR amplification bias and probe affinity, ‘batch effects’ cannot be avoided in gene expression profiling data collected under different noising circumstances including time, platforms, methods, technology and laboratories. The ‘batch effects’ are non-biological technical variations and require to be removed before integrative analysis [14,15]. Recently, SEGs have been well explored and applied in RNA-seq data integration analysis [3,16,17]. Cellular SEGs could also be used as markers of specific cell types, or applied in cell decomposition from bulky gene-expression profiling data for tumor or other diseased tissues to facilitate the translational medicine applications [18]. Cell decomposition analysis, especially for immune cells in tissues, has been performed in cancer studies frequently, based on which the associations are further analyzed with the prognosis and treatment responses.

Endeavors have been made continuously to identify the most comprehensive and stable set of HKGs and SEGs, on which gene expression profiling techniques have large influence, especially for the sensitivity, resolution and repeatability of the results. Serial analysis of gene expression (SAGE) or microarray-based experiments were performed to define HKGs in earlier time, by which both the sensitivity and accuracy were low and the HKG lists were rarely consistent among different studies [19,20]. RNA-seq technology broadened the HKG list significantly [1,20,21,22]. For example, from the RNA-seq data of 16 healthy human tissues, Eisenberg and Levanon identified 3804 HKGs [1]. Until the present, RNA-seq data remained the major choice for the enrichment of the HKG list [22]. Despite the advantages over the probe-based gene expression profiling techniques, in HKG or SEG screening, bulky RNA-seq data still suffer from the problems of limited sample size, within-sample cellular heterogeneity, normalization and integration of data with different resources, etc.

Recently, single-cell sequencing (scRNA-seq) technology has been applied in the detection of human and mouse SEGs [3,17]. Either the individual single-cell library preparation techniques represented by Smart-seq or the droplet based high-throughput scRNA-seq techniques provide the expression levels of individual genes in single cells [23,24]. Compared to bulky RNA-seq data, SEGs could be identified with a higher resolution, i.e., at the single-cell level, from scRNA-seq data, and within-tissue cellular heterogeneity could be avoided. The SEGs identified from scRNA-seq data were more stable than those identified at cell population or tissue levels [3]. Currently, there were limited single-cell SEGs studies, which mainly identified SEGs from Smart-seq scRNA-seq datasets, and encountered the problems in respect to limited cell number, multi-cell data normalization and data integration [3,17]. In contrast, droplet-based high-throughput scRNA-seq techniques could detect gene expression of thousands of cells within a single tissue simultaneously. Non-stochastic gene expression noises among single cells for the same sample are much smaller than those among Smart-seq cells. Therefore, gene expression comparison and its based SEG identification could be performed directly without batch effect removal among the single cells from the same tissue sample in a single droplet-based scRNA-seq experiment.

In this research, with high-throughput scRNA-seq data collected with droplet-based or similar technology, we proposed an iterative strategy to investigate human spatiotemporal SEGs at single-cell level systematically. We also observed the development-, tissue- and cell-specificity of the SEGs, their expression stability under various diseased states, and their applicability as internal controls, cell markers and their based cell decomposition from gene-expression profiling data.

## 2. Results

### 2.1. Spatial-Temporal SEGs among Human Healthy Tissues

A hierarchical procedure was proposed to identify cellular, tissue and general SEGs from high-throughput single-cell RNA-seq datasets generated by 10x Genomics, Drop-seq or Microwell-seq, etc., (Figure 1A; Methods, Section 4.6). For each sample, cells were clustered and identified for their cell types, followed by identification of cluster SEGs, cell-type SEGs, sample SEGs, tissue SEGs and cross-tissue SEGs. Cluster SEGs, cell-type SEGs and sample SEGs were identified according to the Gene Expression Stability Index (GESI) across cells, clusters and cell types (Methods, Section 4.3 and Section 4.4), while tissue SEGs and cross-tissue SEGs were identified according to the distribution of SEGs among samples and tissues, respectively (Figure 1A; Methods, Section 4.5).

SEGs were identified from a list of Microwell-based single-cell RNA-seq data for 44 human adult and fetal tissues involving 97 different samples. 408 SEGs were captured from both at least one adult and one fetal tissue, averagely distributed in nine and more different tissues (Figure 1B; Appendix A). It should be noted that the SEGs might be identified from a few rather than all the different tissues, but they remained spatial since (1) each tissue-SEG was required strictly to be stably expressed among all cell types, and among cell clusters for each cell type, at least in one sample from the corresponding tissue, and (2) there were multi-origin cell types within each sample. In different samples, the SEGs showed invariable expression level among single cells (Figure 1C) and consistently higher GESI than non-SEGs (Figure 1D). Compared to non-SEGs, SEGs always showed higher expression correlation between cell clusters in different samples (Figure 1E; PCC of 0.90 vs. 0.45 for SEGs vs. non-SEGs, *p* < 2 × 10^−16^, Mann-Whitney *U* test). Therefore, the SEGs identified were generally reliable.

Functional enrichment analysis demonstrated that SEGs were enriched significantly in the genes that encode ribosome constituent structure, and participate in transcription, translation and other essential cellular activities (Appendix A). The results are consistent with previous observations and functional expectation on HKGs, suggesting that SEGs and HKGs could show apparent overlap in both the composition of genes and the essential function [1,2].

### 2.2. Comparison with Other SEG Sets

A list of SEGs was identified with the same strategy from 10x Genomics single-cell RNA-seq datasets of human PBMC samples. Generally, there was consistency between the two SEG sets, with ~65% (265/408) of the spatial-temporal (Microwell-seq) SEGs overlapping with the PBMC SEG set (Figure 2A). Many PBMC SEGs could be tissue specific, and therefore were not covered by the Microwell-seq set. The Microwell-seq SEGs were also compared with SEGs identified from Smart-seq single-cell RNA-seq datasets [3] and bulky RNA-seq datasets [1]. There were also ~65% (266/408) of the Microwell-seq SEGs covered by either the Smart-seq or bulky RNA-seqSEGs (Figure 2B). Despite the similar number of Microwell-seq SEGs not covered by the 10x Genomics PBMC set or the Smart-seq/bulky RNA-seq sets, the subsets were not identical in gene composition and therefore not Microwell-seq technique related (Appendix A). When more Microwell-seq SEGs with lowered confidence were included according to a relaxed criterion, i.e., with invariable expression at least in one adult or fetal tissue, the number of overlapping SEGs with the Smart-seq/bulky RNA-seq sets increased extensively but the percentage decreased (645/1269, 51%; Appendix A).

Genes were randomly chosen from the 142 unique Microwell-seq SEGs compared to the Smart-seq/bulky RNA-seq sets. The GESI was consistently high in all the single-cell clusters in the samples selected randomly (Figure 2C). However, in the same samples, randomly selected SEGs from the 317 Smart-seq or 3026 bulky RNA-seq unique subsets did not show high GESI (Figure 2C). Single-cell distribution analysis on the expression of representative genes in the samples also demonstrated the even and high expression of the Microwell-seq unique SEGs but unstable expression of the Smart-seq or bulky RNA-seq unique genes (Figure 2D).

Taken together, the results further demonstrated the reliability of the SEGs identified in this study, and they also broadened the total human SEG list.

### 2.3. Developmental Stage and Cell-Type Specific SEGs

A large variety of SEGs were identified from different tissues, the clustering of which by their presence patterns did not show apparent regularity for the tissues, except for some atypical grouping-together trend for the tissues from the same developmental stages, i.e., adult and fetal stages (Appendix A). Unique SEGs were therefore retrieved for different developmental stages. It was interesting to find that the most reliable adult-stage specific SEGs, which were common in no fewer than five adult tissues but not present in any fetal tissue, were mainly immune-related genes, e.g., *HLA-A*, *HLA-B*, *HLA-C* and *HLA-E* (Figure 3A). The four *HLA* genes are all HKGs identified elsewhere (https://housekeeping.unicamp.br/?visualization (accessed on 12 July 2022)). They were not expressed stably in fetal tissues, potentially suggesting the immatureness of acquired immune systems. It is unclear about the role of *JUN* that was stably expressed in adult but not in fetal tissues. For the fetal stage, spliceosome related genes, such as *SNRPE*, *SNRPF* and *SF3B6*, were apparently enriched in its unique SEGs (Figure 3A). The criteria to identify the development-stage specific SEGs, that is, being common in no fewer than two tissues from one stage and not present in any tissue from the other stage, functional analysis was performed to the stage-specific SEGs, and consistently, the SEGs unique to adult tissues were enriched in immune processes while the SEGs unique to fetal tissues were enriched in RNA splicing significantly (Figure 3B).

Different from tissues, the same or histologically close cell types tended to cluster together according to their SEG presence patterns (Figure 3C). However, the regularity was not very typical and the relationship was not definitive, since developmental stage and tissue were frequently confounding factors that influenced the clustering of cell type, e.g., the cervix cell types, macrophages and endothelial cells of the same developmental stages being grouped, respectively (Figure 3C).

Thirty-two common SEGs were identified from all the cell types with different tissue and development origins (Appendix A). Among them, 26 (81%) encode ribosome proteins (Appendix A). Others include *TMSB10*, *B2M*, *H3F3B*, *TPT1*, *FTH1* and *EEF1A1* (Appendix A). For the SEGs that are shared by most but not all cell types, two Erythroid cell types from fetal heart and skin showed the largest variation, followed by other cell types from fetal and adult heart (Figure 3C; Appendix A). The total number of SEGs identified from these cell types and tissues was also smaller than others generally (Appendix A), as could be due to the lower data quality or the higher heterogeneity for these cells and tissues. In contrast, other cell types, e.g., adult intestine and stomach cells, showed an apparently larger number of SEGs (Figure 3C; Appendix A). Cells from adult intestine and stomach tissues were particularly interesting, for which most SEGs were identified but the profile of SEGs could cluster the cells into two unique sub-groups, epithelium cells and immune cells (Figure 3C). Taken together, despite the possible influence caused by the data quality and number of samples, the results can still partly suggest that cell types, tissue origins and developmental stages showed certain specificity for SEG composition.

### 2.4. Tissue SEGs and Within-Tissue Cell-Type Specific SEGs

In most cases, biomedical studies focus on tissues rather than the whole body. Therefore, it could be more practically useful to identify and apply tissue SEGs and within-tissue cell-type specific SEGs. Generally, 13~472 reliable SEGs were identified for 29 human tissues, which were common across all the cell types in at least one representative sample and were not developmental stage specific (Figure 4A; Appendix A). The variation in SEG number could be due to uneven number of available samples, cellular complexity and heterogeneity within samples, and sample quality. For the batch of tissues analyzed, bone marrow and fallopian tube were identified with the smallest number of cross-cellular SEGs, only 13 and 15, respectively, while thymus was identified with the largest number, 472 (Figure 4A).

For each individual tissue, followed by analysis of the within-tissue cell-type specific SEGs, specific SEGs could be identified for most cell types, and yet only a few of them showed strikingly higher expression level than other cell types so as to be capable of serving as stable markers for the corresponding cells within the tissue (Figure 4B,C; Appendix A).

### 2.5. Stable Expression of SEGs under Disease or Stress Conditions

It remained unclear whether diseases or stress conditions could influence the expression stability of SEGs in different tissues. To answer the question, public single-cell RNA-seq datasets (10x Genomics) were collected that profiled gene expression in different tissues of individuals with various diseases (Appendix A), and SEGs were identified from these diseased tissues.

SEGs were identified from the PBMCs of patients with various tumors, systemic lupus erythematosus (SLE), active tuberculosis (TB) and latent tuberculosis infection (LTBI), and compared with the SEGs analyzed from healthy human PBMC samples, respectively. Except for large cell neuroendocrine carcinoma (LCNC, 67.2%), more than 80% of the SEGs from healthy PBMCs were repeatedly covered by those identified from the PBMCs of patients with all the other diseases (Figure 5A). For TB and LTBI, the SEG coverage was larger than 90% (Figure 5A). SEGs were also identified from T cells of PBMC samples infected with HIV-1 and non-infected controls. Similarly, a high proportion (>90%) of the control SEGs was covered by those identified from HIV-1 infected samples (Figure 5A).

The above diseases may not change the nature but stimulate the gene expression status of PBMCs. To further observe whether cell composition changes caused by diseases could influence the SEGs, solid samples with various diseases, especially tumors, were analyzed for the tissue SEGs, and compared with the corresponding SEGs identified from healthy tissues (Figure 5B). We noticed that the SEGs identified from various diseased tissues were more than those from healthy tissues. This could be due to the larger number and higher data quality for the diseased samples included for analysis, although the possibility cannot be excluded that more genes are activated in the pathologic state. Similarly, in the PBMC samples, for all the diseases of solid tissues, a majority (>80%) of the SEGs of healthy tissues were covered by the SEGs identified from the diseased tissues (Figure 5B).

Taken together, the results demonstrated that most SEGs identified from healthy tissues also expressed stably under diseases or stress conditions.

### 2.6. Application of SEGs in Cell Decomposition of Samples

SEGs have multiple applications, for instance, serving as cell markers, being used as internal control for gene expression comparison of target genes from different tissues, gene expression normalization and data integration, etc. Here, taking PBMC SEGs as an example, we tested their possible application in cell decomposition of multi-cellular samples based on gene expression profiling data.

Cell-specific SEGs from healthy PBMC samples for four major immune cell types, i.e., T, B, Myeloid (M) and Natural Killer (NK) cells, were used for deconvolution model training with the 3K PBMC dataset from 10x Genomics website. The training dataset and other independent PBMC single-cell RNA-seq datasets with known composition of major immune cells, including the 10x Genomics 10K dataset, another healthy PBMC dataset (HC1), an LTBI PBMC dataset (LTBI1) and a TB PBMC dataset (TB), were used for assessment of the model performance. As shown in Figure 6, the cell deconvolution model, SEGdecon, could predict the composition percentages of the major immune cell types at an accuracy of > 0.86 for all the testing datasets. It should be noted that the NK/T cells are difficult to separate from the scRNA-seq data of TB and LTBI samples, and therefore merged into a unique cell type [27]. However, the prediction model could still predict the percentages of NK and T cells separately (Figure 6).

## 3. Discussion

The notion of an HKG has been ambiguous since the concept was proposed. The ambiguity stemmed from both the basic definition and the approaches to identify them. An HKG could be understood as a gene with either universal or invariant expression within all the tissues at different developmental stages for a specific multi-organ organism, and consequently the strategies for identification are different. Traditionally, a ‘tissue-specifically expressed gene’ (TSG) is a notion in contrast to an HKG, strengthening the heterogeneous expression among different tissues. When HKGs or TSGs are detected from tissue level, a TSG could be a gene with varied or stable expression among different samples of the same tissue but with varied expression among different tissues. At single-cell level, however, traditional TSGs could be defined more clearly with three groups: (1) tissue-specific SEGs with invariant expression among different cells of the same tissue but with unstable or zero expression for all the other tissues of an organism, (2) tissue SEGs with invariant expression among different cells of the same tissue but not with stable expression for all different tissues of an organism, and (3) tissue-specific genes that are specifically expressed with invariant or varied levels in some of the cells within a tissue. Correspondingly, HKGs can be described as global cellular SEGs more definitively, i.e., genes with invariant expression among all the cells of an organism. An HKG (defined by invariant expression) screened from tissue level could be re-classified as a TSG through scRNA-seq analysis, since the HKGs should meet with much stricter criteria, especially cross-cellular stableness within a tissue. At the tissue level, the major types of composition cells have a dominant influence on the quantification of genes, and therefore within-tissue gene expression levels tend to represent them in the major cell types. At the single-cell level, however, taking the current study as an example, we normalized the influence of various cell types, especially strengthening the influence of relatively rare cell types. Therefore, similar to other studies [3,17], it is not surprising that the global SEG list identified in the study is much smaller than those identified from bulky RNA-seq datasets. Droplet-based scRNA-seq data captured much more single cells and cell types within individual samples than Smart-seq scRNA-seq data, while the iterative SEG identification strategy adopted in this study emphasized the accuracy with a tradeoff of sensitivity, and consequently we identified fewer global SEGs (408) than those identified from Smart-seq data (1076) [3,17]. In fact, considering the composition of multiple cell types of different origins in each tissue, a majority of the union set of tissue SEGs could potentially represent real global ones, and the HKG list would be extended to contain 1269 genes, more than those identified from Smart-seq data (Appendix A). However, we still recommend usage of the stricter ones since the latter are more stable. In addition to scRNA-seq data quality and cell number within each dataset, the number of global SEGs is dependent on the size of samples for individual tissues and tissue types before saturation. As more high-quality scRNA-seq datasets are available, they can be integrated to enlarge the size of global SEGs. We have initiated a project to update the list of human global SEGs regularly (http://61.160.194.165:3080/hSEGdb, accessed on 12 July 2022). Meanwhile, it is also interesting to investigate how many tissues and samples ensure the saturation of SEGs to be identified.

The HKG lists often showed low consistency among different studies [1,3]. It could be due to the difference in HKG notion as described above, resolution (tissue or cell level), types and number of included samples, tissues or cells, analytic methodology, etc. The global SEGs identified from this study also showed certain non-overlapping percentages with other HKG lists (Figure 2B). However, regarding observations on the expression of multiple inter-consistent genes within different samples and tissues, we demonstrated that our global SEGs appeared more stable and reliable (Figure 2C,D). Similar to the number of identified SEGs, the stableness of global SEGs is also dependent with the size of samples, the diversity of tissues, the number and types of cells, and the quality of scRNA-seq data. As more diverse and high-quality scRNA-seq datasets are included, more global SEGs with higher confidence could be identified. The algorithms and computational methods could also influence the reliability of resulted global SEGs. In this research, an optimized hierarchical strategy was applied with strict criteria at the cell cluster and cell type levels, and relaxed, distribution-based inter-sample and inter-tissue SEG analysis according to the currently available size of subjects that were analyzed (Figure 1A). The strategy is also sample/tissue size dependent, and more reliable SEGs will be found when more high-quality scRNA-seq datasets are included in future.

In addition to global SEGs, the study also investigated development-, tissue- and cell-specificity of SEGs among or within human tissues. First, we found that the developmental stages of human tissues showed specific SEGs, and that SEGs specific to adult tissues were enriched in immune functions while SEGs specific to fetal tissues were enriched in RNA splicing activities (Figure 3A,B). Despite this being the first report to our knowledge, the observations are expected theoretically. The acquired immune systems remain immature in fetuses, though some essential immune-related genes are important and should be expressed stably in multiple, even immune-unrelated, adult tissues [28] (Figure 3A). In contrast, the fetal tissues have more developmental potential and alternative splicing is a relatively universal process for developmental regulation. The expression of the splicing factors could show larger specificity in more differentiated adult tissues [29]. Therefore, it is expected that more genes encoding splicing factors show stable expression within fetal tissues. In this study, the SEGs showed specificity and clustering patterns for cell types but not for tissues (Figure 3C; Appendix A), as were also normal, since the samples of each individual tissue contained multiple cell types of various developmental origins and most differences among tissue SEGs were caused by non-biological systematic noises. To better characterize and apply SEGs, we proposed the notions of ‘tissue SEGs’ and ‘within-tissue cell-specific SEGs’ (Figure 4A,B), which made different tissues as units and screened the common SEGs among all the cell types and cell-type-specific SEGs within each individual tissue, respectively. Tissue SEGs and within-tissue cell-specific SEGs are of more practical use than global SEGs and cell-specific SEGs in that the number of the former is larger, the robustness is higher, and a specific tissue was focused more occasionally than the whole human body in real studies. Another important finding of the study is the demonstration of rare influence of diseases or stresses on the expression stableness of SEGs (Figure 5). Although the types of diseases or stresses for each tissue were limited, the results remained reliable since the disease or stress types for all tissues were diverse, and there was a large consistency among multiple tissues (Figure 5). The finding laid the foundation for the various applications of SEGs, for instances, data normalization and integration, cell decomposition, etc.

SEGs have many potential applications. In this study, we tested their possible applications as cellular markers and for cell decomposition. Within-tissue cell-specific SEGs could be divided into two types, marker SEGs that are not or much more lowly expressed in other cell types, and the remainder. Only the former could serve as effective markers for specific cells within a type of tissue. We found that most of the cell types within individual tissues could be identified with an extensively reduced number of marker SEGs (Figure 4B). These genes could be well used as the markers for specific cells (Figure 4C). It should be noted that the prior knowledge on cell types and single-cell clustering are important for the identification of within-tissue cell-specific SEGs and the corresponding marker SEGs. The hSEGdb database also curated and will regularly update the tissue SEGs, within-tissue cell-specific SEGs and marker SEGs for various human tissues, with continuous inclusion of more scRNA-seq datasets and more thorough prior knowledge on the tissues and cell composition. We also briefly examined the possible application of within-tissue cell-specific SEGs in cell decomposition for bulky gene-expression profiling data. The PBMC samples were tested, for which only four major immune cell types were decomposed, from semi-supervised datasets with known composition and percentages of each type of cells. The simple model based on a set of over determined equations taking within-tissue cell-specific SEGs as variables predict the composition of different cells with an accuracy of >0.86 for each independent testing dataset (Figure 6). The results suggested the potential application of the SEGs, though more elaborate models need to be trained for more tissues with more specific cell types being decomposed, and with more comprehensive performance assessment especially in real bulky RNA-seq or microarray datasets.

Serving as inner control genes for quantitative RT-PCR (qRT-PCR) could be another important application of SEGs. However, the commercial set of human common HKGs, which are also frequently used as qRT-PCR inner controls in laboratory experiments, are often not among the SEG list we identified at single-cell resolution. For example, six among the fifteen endogenous reference genes of Applied Biosystems (http://www.appliedbiosystems.com, accessed on 12 July 2022) were not covered by the global SEGs or tissue-SEGs, including *GUSB*, *HPRT1*, *TBP*, *IPO8*, *POLR2A* and *HMBS*. Genes identified as reference genes for qRT-PCR normalization in specific tissues are also frequently recognized as not stably expressed in this research. For instances, *SDHA* and *TBP* in human bladder cancers [30], *CYC1* and *TOP1* in human brains [31], and *TRAP1*, *FPGS* and *DECR1* in human PBMC samples [32], were identified as tissue HKGs that could be used as endogenous control genes, but none of them is with stable expression in corresponding tissues or cells analyzed in this study. The genes listed here are not necessarily non-SEGs, but did show larger variance than other SEGs in the included samples. In practice, 32 spatial-temporal SEGs that are repeatedly identified from all the cell types should be more reliable and used as the first choices for qRT-PCR experiments (Appendix A, Cell_type_SEGs, the top 32 genes). More emphasis should be placed in validation of more reliable SEGs in different normal and diseased tissues. Except for cell decomposition and endogenous reference genes in qRT-PCR experiments, other SEG-based applications were not tested in the study but examined elsewhere, for example, single-cell data normalization and integration [3,17]. It would be also interesting to observe whether the global SEGs and tissue SEGs identified in this study have similar or better effect in the same applications.

In summary, the study identified a list of human reliable global SEGs at single-cell resolution, found the development- and cell-specific SEGs, demonstrated rare or no influence of diseases or stresses on SEGs, and examined the possible applications of the SEGs. There are reports that demonstrated different expression patterns between highly and lowly expressed genes in single cells [33,34]. Moreover, many genes show zero expression in single cells [35,36]. The GESI-based SEG identification pipeline is more sensible for the highly expressed genes. Therefore, there is a potentially large space to add the SEG list, and more endeavors should be placed in clarification of the expression distributions of genes and identification of the lowly expressed SEGs in future. The SEG list should also be updated with more high-quality scRNA-seq data. Despite the large consistency of SEGs between normal and diseased tissues, the number in the latter appeared larger apparently (Figure 5). The larger number and higher cell and data quality of diseased samples could be important factors, and yet more work is required to address the question—whether more genes being activated causes more SEGs being captured. For each type of tissue, more refined algorithms are needed to further ensure that the SEGs from diseased and normal samples are with the same level of expression. While the current study only observed the stable expression of genes at the single-cell level, recent new techniques have been developed that could identify and quantify the expression of single-cell isoforms and even allele-specific isoforms [25,37,38,39]. Therefore, it would also be interesting to further the single-cell stable expression analysis on transcripts and allele-specific isoforms.

## 4. Materials and Methods

### 4.1. Single-Cell RNA-Seq Datasets and Data Preprocessing

Microwell-seq based single-cell RNA-seq datasets were generated for more than 50 adult and fetal human tissues [40], which were used for hSEG identification in this research. Other 10x chromium single-cell RNA-seq datasets for various healthy or diseased human tissues were downloaded from NCBI Gene Expression Omnibus (GEO) database or Panglaodb [41] (Appendix A).

For the single-cell RNA-seq data of each individual sample, the R package, ‘Seurat’ (version 4.0.2), was used for data preprocessing [42]. Quality control such as doublet detection and cell filtering was performed according to the original processing protocol for the sample recorded in the corresponding published paper. Identification of the most variable gene signatures, principle component analysis (PCA) and clustering were performed with the routine ‘Seurat’ protocol [42]. Cell type identification for each cluster was according to the putative cell markers or the information annotated from the original reports.

### 4.2. Binning of Single Cells

The amount of gene transcripts in individual single cells, as measured by Unique Molecular Identifiers (UMIs), is too small and the variance is large caused by random sampling errors rather than real heterogeneity. To overcome the influence of such random errors and also to retain the real heterogeneity to the best, *n* cells within each homogenous cluster were binned and the UMIs for each gene within the cells of each bin were combined. The clusters with < *m* cells within each sample were excluded for gene expression stableness analysis, where *m* represented the minimal number of total cells in each cluster that was used for further analysis. Within each cluster, resampling was performed to form the cell bins for *r* rounds. For each round, *n* cells were extracted randomly from the whole set to form an individual bin. In this research, *n*, *m* and *r* were optimized according to the number of cell clusters that can be used for further analysis, the overall stableness of gene expression among bins and the power of SEGs identified from the cell clusters, and eventually set as 10, 100 and 100, respectively.

### 4.3. Gene-Expression Stableness Index

Gene-expression stableness index (GESI) is defined as $\frac{1}{1+\delta /u}$ where δ  and u mean the standard deviation and mean of gene expression among single-cell bins or clusters, respectively. GESI ranges from 0 when the variance is very large, to 1 if the variance is zero. A higher GESI means that expression of the gene is more stable.

### 4.4. Inter-Cluster Analysis of Gene Expression Stableness

Within each sample, the mean and standard error of the expression mean of all the bins in each cluster were calculated. GESI was further calculated for each gene among the clusters. Due to the large noises of single-cell RNA-seq data, three factors should be considered when SEGs are identified among cell clusters, including the reliability of SEGs within clusters, the recurrence of SEGs among clusters, and the inter-cluster expression stableness of the SEGs. To balance the power and precision, we tested the different combination of the parameters using a 10K PBMC scRNA-seq dataset from the 10x Genomics website, i.e., top 100, 500 and 1000 stably expressed genes within clusters, common within-cluster SEGs among 50%, 75% and 100% clusters, and an inter-cluster GESI cutoff of 0.667, 0.714, 0.769, 0.833 and 0.909. Finally, inter-cluster SEGs are defined that should meet the following optimized criteria: (1) top 1000 genes with largest GESIs within each cluster (within-cluster SEGs); (2) common within-cluster SEGs among all clusters; (3) with an inter-cluster GESI larger than 0.667 (i.e., δ/u < 0.5).

### 4.5. Stableness Analysis of Gene Expression between Samples

The variance of gene expression from cell to cell could be sourced from either real biological heterogeneity or stochastic errors, or both. One objective in screening SEGs is to distinguish and remove the genes with varied expression caused by biological heterogeneity. Recurrence should be an important measure to reduce the influences caused by random errors, especially for the small amounts of transcripts in single cells. General expression collinearity for multiple genes is another important indicator for measuring stableness between samples, especially for those derived from different tissue origins. Pearson correlation coefficients were calculated and compared between samples for SEGs or non-SEGs.

### 4.6. A Hierarchical Pipeline for SEG Identification

A hierarchical pipeline was proposed for the identification of SEGs at the single-cell level (Figure 1A). It was divided into six steps. (1) Single cell clustering and cell type identification. After preprocessing, the scRNA-seq matrix for each sample was loaded to R, and clustered with Seurat or other tools. The cell type was identified for each cluster, according to the expression and distribution of putative cellular markers or the information provided in the dataset-corresponding reference publications. (2) Identification of within-cluster SEGs. For each cluster containing ≥ 100 cells, cells were binned, GESIs were calculated, and within-cluster SEGs were retrieved (1000 with the largest GESIs for each cluster) according to the corresponding procedure described in the above methodology sections. (3) Identification of inter-cluster SEGs for each cell type. The union set of within-cluster SEGs and their cluster distribution were analyzed for each cell type of a sample. The SEGs shared by most clusters were retained, and the total number of retained SEGs was set as not smaller than 500 in this research. For each of the retained SEGs, inter-cluster GESI was calculated. A gene was recognized as a cell type SEG if the inter-cluster GESI was larger than 0.667. (4) Identification of sample SEGs. The intersection of cell type SEGs was analyzed for each sample, and the GESIs among all the cell types were calculated. A gene was recognized as a sample SEG if the gene was shared by all the cell type SEG lists and the GESI among the cell types was larger than 0.667. (5) Identification of tissue SEGs. The union set of sample SEGs and their distribution were analyzed for each tissue. A cutoff *c* was set and the SEGs shared by ≥ *c* samples of the corresponding tissue were retained and identified as tissue SEGs. Despite SEGs with larger stableness for a larger *c*, the number of samples for most tissues in the study was limited. Consequently, *c* was set as 2 if there were more than one sample for the tissue, 1 if there was only one sample. (6) Identification of global (cross-tissue) SEGs. Similar to the last step, the union set of tissue SEGs and their distribution were analyzed. A global SEG was defined if it was covered by at least *l* different tissues.

### 4.7. Detection of Within-Tissue Cell-Specific SEGs and Marker SEGs

Cell type composition was annotated for each tissue, followed by the identification of within-tissue SEGs for each cell type. A within-tissue cell-specific SEG was both defined and identified only if it was a SEG unique for one cell type within the tissue. Once the cell-specific SEGs were identified, their average expression in the single cells of corresponding cell type (target) and other cell types was calculated and compared for each sample of the tissue. The marker SEGs were defined as the subset of within-tissue cell-specific SEGs with (1) more than 2-fold increased expression in the target cells compared to each of the other cell types, and (2) significant expression difference between the target and the other cell types (one-side students’ T-test, *p* values < 0.05).

### 4.8. Cell Decomposition Models

The 3K and 10K PBMC scRNA-seq datasets were downloaded from the 10x Genomics website. Cells were clustered, and T-, B-, NK- and M-cells were identified according to the putative marker genes for corresponding cell types. Other 10x Genomics PBMC scRNA-seq datasets including one from healthy subject (HC1), one from a TB patient (TB1) and another from a LTBI patient (LTBI1) were downloaded from NCBI GEO database, and the M, B and NK/T cells were annotated according to the original reference [27]. Cell-specific SEGs of healthy PBMC tissues were used as variables for training the cell deconvolution model (SEGdecon) based on a set of overdetermined equations, and the 3K dataset was used as the training dataset. Suppose there were *s* cell-specific SEGs. For each testing dataset, *s* equations could be generated as
$$\left[\begin{array}{c} \underset{c=T,B,NK,M}{\sum }w1_{c}\;*\;f_{c}=E1\;\left(1\right)\;\\ \underset{c=T,B,NK,M}{\sum }w2_{c}\;*\;f_{c}=E2\;\left(2\right)\\ \ldots \ldots \\ \underset{c=T,B,NK,M}{\sum }ws_{c}\;*\;f_{c}=Es\;\left(s\right) \end{array}\right]$$
where *c* represented an immune cell type from the space of [T, B, NK and M], *ws_c_* was the coefficient parameter and represented by the average expression level of the *s*^th^ cell-specific SEG in cell type *c* of the training dataset, *f_c_* represented the fraction of cell type *c* in the testing dataset, and *E_s_* was the general expression of the *s*^th^ cell-specific SEG in the testing dataset calculated as the average UMIs among all the analyzed single cells. For each SEG, the 4 *w*’s were determined with the training dataset, which were represented by the average expression in corresponding cell types. The *f_c_*’s were unknown and need to be predicted for each testing dataset. Since the number of *f_c_*’s, 4, was smaller than the number of equations, *s*, there were no definite solutions for *f_c_*’s. Once the *ws_c_*’s were determined, the optimized solutions of *f_c_*’s for each testing dataset were inferred with a least squares (LS) method. The solution of overdetermined equations was performed in R. For each testing dataset, both the cell type identity for each single cell and the real fraction of each cell type were known. For each cell type, the prediction accuracy was defined as: *1- absolute (real fraction—predicted fraction)*/*real fraction*. The average accuracy was calculated to evaluate the performance of SEGdecon models for each testing dataset.

### 4.9. Codes Availability

The scripts assisting identification of SEGs and usage of the SEGdecon model, together with the documents and SEG datasets, have been stored in the website and can be downloaded freely: http://61.160.194.165:3080/hSEGdb, accessed on 12 July 2022.

### 4.10. Statistics

Statistic tests involved in the study were indicated in context. R version 3.3.3 was used for the statistical analysis. The significance level (alpha) was preset as 0.05.

## Figures and Tables

**Figure 1 ijms-23-10214-f001:**
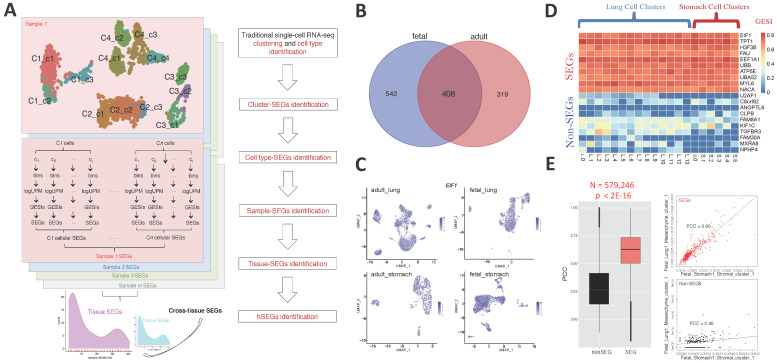
Identification of SEGs at single-cell level. (**A**) The pipeline for SEG identification. (**B**) 408 spatial-temporal SEGs were identified from both adult and fetal tissues. (**C**) Expression of a representative spatial-temporal SEG, *EIF1*, in single-cell clusters of representative samples. One sample was selected randomly for each of the adult lung, adult stomach, fetal lung and fetal stomach tissues, respectively [25]. The single cells were clustered according to the gene expression profile, and shown as points in a two-dimension plane of Uniform Manifold Approximation and Projection (UMAP) [26]. The gene expression level was represented by the color shown in the scale bar. (**D**) GESI heatmap of representative SEGs and non-SEGs in cell clusters of representative samples. Ten spatial-temporal SEGs and ten non-SEGs were randomly selected. Two scRNA-seq samples were selected from the adult tissues randomly [25]. (**E**) Expression correlation of SEGs and non-SEGs between samples. All 408 SEGs and paired 408 non-SEGs randomly selected from the non-SEG gene set were used for the analysis and comparison. The Pearson Correlation Coefficients (PCCs) were calculated for SEGs and non-SEGs between each inter-sample cell-cluster pair, respectively, and there were 579,246 pairs in total. A Mann-Whitney *U* test was performed between the PCCs of SEGs and non-SEGs. The correlation of SEGs and non-SEGs between a representative cell-cluster pair was also shown.

**Figure 2 ijms-23-10214-f002:**
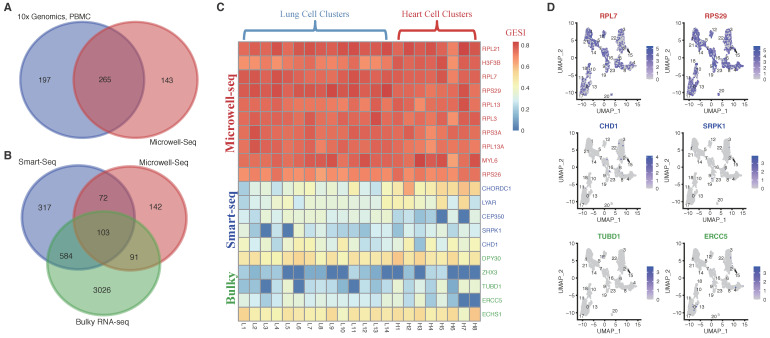
Comparison of SEGs identified from different datasets. (**A**) Comparison of the spatial-temporal SEGs (Microwell-Seq) and PBMC SEGs identified from 10x Genomics scRNA-Seq datasets. (**B**) Comparison of the Microwell-Seq SEG, Smart-Seq SEG [3] and bulky RNA-seq HKG lists [1]. (**C**) GESI heatmap of representative genes inconsistent among the Microwell-Seq SEG, Smart-Seq SEG and bulky RNA-seq HKG lists. Ten Microwell-Seq SEGs, five Smart-Seq SEGs and five bulky RNA-seq HKGs not covered by the other two gene lists (from the specific subsets shown in Figure 2B) were selected randomly, and their cellcluster GESIs in two representative samples were shown. The samples were randomly selected from [25], both of which were from adult tissues. (**D**) Expression of two representative SEGs/HKGs specifically detected by Microwell-Seq (*RPL7* and *RPS29*), Smart-Seq (*CHD1* and *SRPK1*) or bulky RNA-seq (*TUBD1* and *ERCC5*) in single-cell clusters of a representative sample. The representative sample (an adult lung sample) was selected from [25] randomly.

**Figure 3 ijms-23-10214-f003:**
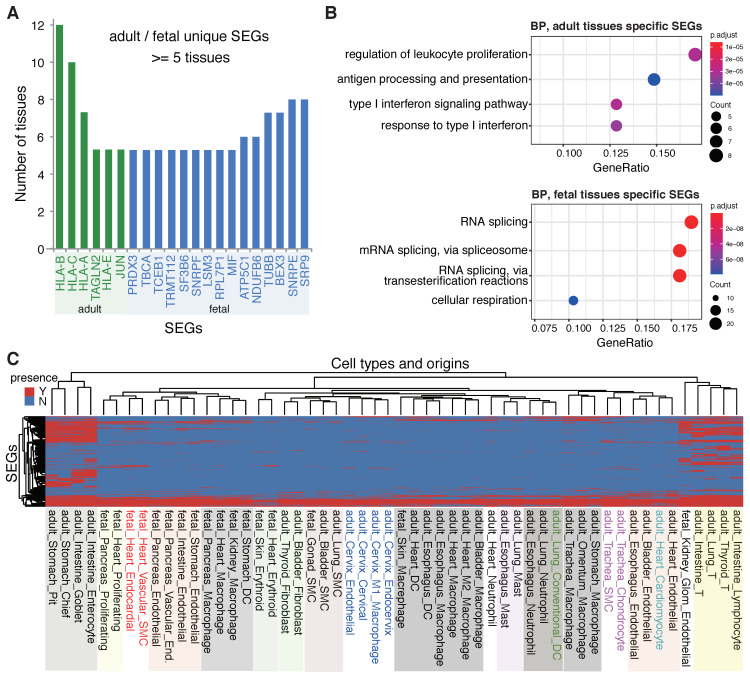
Development- and cell-specific SEGs. (**A**) SEGs specific to human adult tissues and fetal tissues. The adult-specific SEGs were identified from at least five adult tissues but none of the fetal tissues, and vice versa for the fetal-specific SEGs. (**B**) Functional enrichment for the development-specific SEGs. For the enrichment analysis, the adult-specific SEGs were identified from at least two adult tissues but none of the fetal tissues, and vice versa for the fetal-specific SEGs. Gene ontology biological process (BP) terms were used and Fisher exact tests with Bonferroni correction were performed. (**C**) Clustering of within-tissue cell-type SEGs. Genes and cell types were represented in rows and columns, respectively. The value for the heatmap cells was set as ‘Y’ (red) if the SEG was present for the corresponding cell type, and ‘N’ (blue) otherwise. Cell types clustered apparently were shown in colored rectangles.

**Figure 4 ijms-23-10214-f004:**
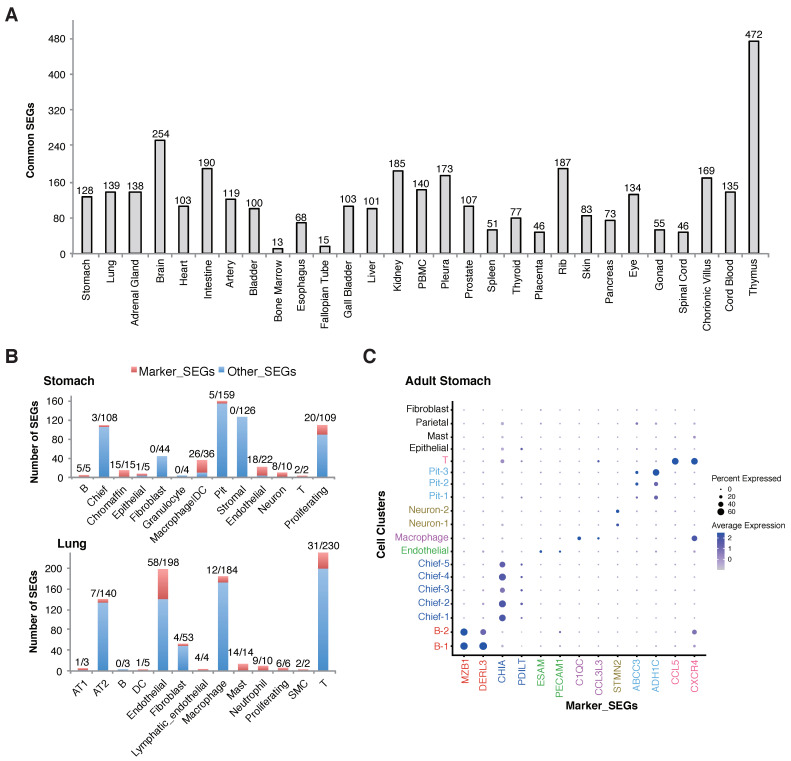
Tissue SEGs and within-tissue cell-specific SEGs. (**A**) Tissue SEGs. The SEGs were common for each cell type within the corresponding tissue. (**B**) Within-tissue cell-specific SEGs and marker SEGs in representative tissues. For each tissue, the composition cell types, the number of cell-specific SEGs, and the number of cell-specific marker SEGs were shown. For the numbers presented as ‘x/y’ on the top of each bar, ‘x’ and ‘y’ represented the number of cell-specific SEGs and marker SEGs, respectively. Marker SEGs represent SEGs that are expressed specifically or with striking higher level in indicated cell types and can be used as gene markers for the corresponding cells. (**C**) Expression of representative cell-specific marker SEGs in cell clusters of a representative sample. The representative stomach sample was selected from [25] randomly.

**Figure 5 ijms-23-10214-f005:**
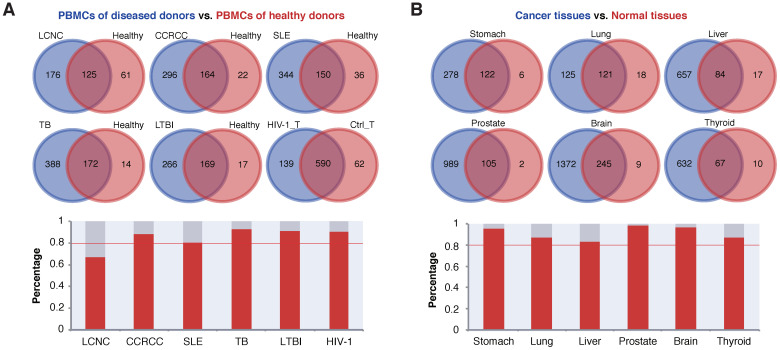
SEGs from tissues under stresses or with diseases. (**A**) Comparison of SEGs from PBMCs (or T cells for HIV-1 infection and control) of healthy donors and patients with diseases. (**B**) Comparison of SEGs from different healthy and diseased solid tissues. For both (**A**,**B**), the percentages of common SEGs (intersects of the Venn graphs) in healthy tissues were shown in the bar plots.

**Figure 6 ijms-23-10214-f006:**
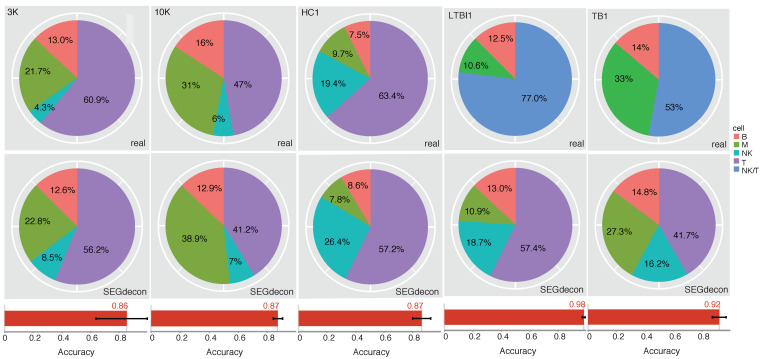
Performance of the SEGdecon model. SEGdcon was used to predict the proportion of major immune cells in PBMC samples. Each testing dataset was shown in a column. The real and predicted composition percentages of each major immune cell type were shown in the pie charts. The average accuracy was shown at the bottom for each testing dataset.

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
