# Peer review of "Identification of Human Global, Tissue and Within-Tissue Cell-Specific Stably Expressed Genes at Single-Cell Resolution"

_ijms, 2022, doi:10.3390/ijms231810214_

Round 1

Reviewer 1 Report (Previous Reviewer 3)

Authors have answered and modified the requested comments I had.

Author Response

Thanks for your comments.

Reviewer 2 Report (New Reviewer)

A very interesting article, the main question of which is related to the use of the terms Stably Expressed Gene (SEG) and Housekeeping Gene (HKG). In fact, the authors propose to redefine the term HKG at the level of single cell transcriptome analysis, thus coming to the phenomenon of multiple HKGs characteristic of different cell types, tissues, developmental stages, and so on. As a result, the term HKG itself is blurred and, apparently, should go into the option of neuronal HKG or muscle HKG.

The main issue for the article is the lack of a concept of the level of expression of a particular gene, expressed through the number of reads. It is not clear whether all genes identified during the sequencing process were taken into account in the search and analysis of SEG - although it is clear that weakly expressed genes with a small number of reads may behave differently compared to actively expressed genes. This may affect the analysis of the Gene-expression stableness index (GESI) . With a small number of reads, estimates of both the mean and variance may be less accurate due to the greater spread of data between different cells. It is also not clear what levels of GESI the authors consider to be threshold for classifying genes in the SEG group.

Data from different single cell sequencing platforms need to be compared in more detail with a more complete overview of the data. For example. Figure 2 A-C - How were the genes selected for comparison in Figure C, taking into account the Venn diagram data in Figures A and B?

Figure 3, SEGs specific to human adult tissues and fetal tissues. The adult-specific SEGs were identified from at least 5 adult tissues - but this is a very strange set of SEGs. It consists of the HLA genes and the immediately early JUN gene. I would like to see an explanation of this in the article.

Figure 5, pathology-norm comparison. The presence of a very large number of SEGs characteristic of pathology is noteworthy - there are more of them than SEGs in normal tissue. But what happens to the expression level of SEG, common for normal and pathological tissue - does it change in pathological tissue, while remaining stable? And does not the increase in the number of SEGs in the pathological tissue just reflect the activation of gene expression with the transition to a higher GESI value.

In general, it should be noted that it is necessary to expand the captions to the figures, they are not very informative and do not explain them. For example, Figure 5 - “The percentages of SEGs from healthy tissues covered by those from tissues under stresses or with diseases were shown in the bar plots." It is extremely difficult to understand how the bar plot was built after all

Author Response

Comments and Suggestions for Authors

A very interesting article, the main question of which is related to the use of the terms Stably Expressed Gene (SEG) and Housekeeping Gene (HKG). In fact, the authors propose to redefine the term HKG at the level of single cell transcriptome analysis, thus coming to the phenomenon of multiple HKGs characteristic of different cell types, tissues, developmental stages, and so on. As a result, the term HKG itself is blurred and, apparently, should go into the option of neuronal HKG or muscle HKG.

Response: Thanks for your comments and the summary of our research design / main findings. Also many thanks for your questions below, which give us a lot of important clues and inspiration and help us improve the manuscript significantly.

The main issue for the article is the lack of a concept of the level of expression of a particular gene, expressed through the number of reads. It is not clear whether all genes identified during the sequencing process were taken into account in the search and analysis of SEG - although it is clear that weakly expressed genes with a small number of reads may behave differently compared to actively expressed genes. This may affect the analysis of the Gene-expression stableness index (GESI). With a small number of reads, estimates of both the mean and variance may be less accurate due to the greater spread of data between different cells. It is also not clear what levels of GESI the authors consider to be threshold for classifying genes in the SEG group.

Response: Many thanks for pointing out this major concern. We adopted a fixed GESI cutoff rather than expression level specific GESI cutoffs to identify SEGs, as performed to bulky RNA-seq or microarray-based gene expression data previously (Trends Genet, 2013, 29:569-74; Nucleic Acids Res, 2021, 49: D947-55). However, the expression patterns and distributions in single cells could potentially be different between the genes with high and low expression levels (Nature, 2013, 498:236-40; Nat Biotechnol, 2013, 31:748-52). The strategy in our study is more sensible to identify the highly expressed SEGs, which are also more accurate and stable. Therefore, the identified list of SEGs is plausible but not comprehensive. For the potentially under-detected SEGs with low expression, the GESI-based strategy could be not suitable, and new methods are required, which should be based on more investigations on the expression patterns and distributions of such genes. We are working on updating and refining the SEG list by including new detection methods and more high-quality single-cell RNA-seq datasets. We also added comments on this possible drawback in the last DISCUSSION section of the revised manuscript: “There are reports that demonstrated different expression patterns between highly and lowly expressed genes in single cells [39-40]. Moreover, many genes show zero expression in single cells [41-42]. The GESI-based SEG identification pipeline is more sensible for the highly expressed genes. Therefore, there is a potentially large space to add the SEG list, and more endeavors should be placed in clarification of the expression distributions of genes and identification of the lowly expressed SEGs in future. The SEG list should also be updated with more high-quality scRNA-seq data”.  

Data from different single cell sequencing platforms need to be compared in more detail with a more complete overview of the data. For example. Figure 2 A-C - How were the genes selected for comparison in Figure C, taking into account the Venn diagram data in Figures A and B?

Response: The genes compared in Figure C were selected randomly from the specific subsets shown in the Venn diagram of Figure B. We have clarified it in the legend of Figure 2 in the revised manuscript.

Figure 3, SEGs specific to human adult tissues and fetal tissues. The adult-specific SEGs were identified from at least 5 adult tissues - but this is a very strange set of SEGs. It consists of the HLA genes and the immediately early JUN gene. I would like to see an explanation of this in the article.

Response: The four HLA genes, including HLA-A, HLA-B, HLA-C and HLA-E, were also identified as house keeping genes elsewhere (https://housekeeping.unicamp.br/?visualization). Besides them, there are also other immune-related genes detected with stable expression in adult human tissues (the threshold being loosen to >= 2 tissues) but not stable or undetectable in fetal tissues. It could potentially suggest the essential function in adults but immatureness of the acquired immune system in fetuses. We are not sure about the role of JUN. Some other studies suggested that JUN could be a stress-induced gene that is frequently detected in single-cell RNA-seq data. However, we found that it is not only frequently detected but with stable expression in multiple human adult tissues, but it is undetectable or not stably expressed in fetal tissues. Therefore, it seemed not to be a ‘biased’ gene and we did not remove it from the SEG list.

In the revised manuscript, we added some explanation on the observation of the development-specific genes in the context of results: “The four HLA genes are all HKGs identified elsewhere (https://housekeeping.unicamp.br/?visualization). They were not expressed stably in fetal tissues, potentially suggesting the immatureness of acquired immune systems. It is unclear for the role of JUN that was stably expressed in adult but not in fetal tissues.” In DISCUSSION (the 3rd paragraph), we also highlighted the brief explanation on the observation of enrichment of immune-related SEGs in human adult tissues but not in fetal tissues.

Figure 5, pathology-norm comparison. The presence of a very large number of SEGs characteristic of pathology is noteworthy - there are more of them than SEGs in normal tissue. But what happens to the expression level of SEG, common for normal and pathological tissue - does it change in pathological tissue, while remaining stable? And does not the increase in the number of SEGs in the pathological tissue just reflect the activation of gene expression with the transition to a higher GESI value.

Response: Thanks for the comments on the different number of SEGs between pathologic and healthy tissues. They are interesting and insightful. We took the SEGs identified from the PBMC T cells with or without HIV-1 infection as an example, and made the following analysis during the revision: (1) For the 590 common SEGs (Figure 5A), we calculated the average expression level, i.e., the mean of UPMs per cell, in the HIV-1 positive samples and the HIV-1 negative samples. The ratios of gene expression between the two types of samples are close to 1, and no enrichment of SEGs with higher expression in HIV-1 positive samples was found (284 higher vs. 297 lower; 9 equal). (2) For the SEGs specifically detected from HIV-1 positive samples, we did not find their significantly increased expression in HIV-1 positive samples compared to the HIV-1 negative samples (none of the SEGs was consistent with the significantly up-regulated ones in HIV-1 positive samples identified through cell-pooled expression level comparison). We also performed the analysis to normal Stomach and gastric cancer samples, and got the similar observation. However, more tissues and samples should be analyzed before the conclusion can be drawn. The current strategy for cross-sample comparison is too rough and simple, which needs to be refined. Some other factors could also have contributed to the observation, for example, the number of pathologic samples frequently larger than that of normal samples, the relatively lower cell and data quality for the normal tissues, etc. We think this could be an interesting subsequent project, which deserves further comprehensive investigation and could hardly be completed during the short revision period.

Therefore, we did not add the preliminary analysis results in the manuscript, but revised the sentences in Section 2.5 to “We noticed that the SEGs identified from various diseased tissues were more than those from healthy tissues. This could be due to the larger number and higher data quality for the diseased samples included for analysis, although the possibility cannot be excluded that more genes are activated in the pathologic state”. In the last paragraph of DISCUSSION, we also added comments accordingly, “Despite the large consistency of SEGs between normal and diseased tissues, the number in the latter appeared larger apparently (Fig. 5). The larger number and higher cell and data quality of diseased samples could be important factors, and yet more work is required to address the question - whether more genes being activated causes more SEGs being captured. For each type of tissue, more refined algorithms are needed to further ensure that the SEGs from diseased and normal samples are with the same level of expression”.

In general, it should be noted that it is necessary to expand the captions to the figures, they are not very informative and do not explain them. For example, Figure 5 - “The percentages of SEGs from healthy tissues covered by those from tissues under stresses or with diseases were shown in the bar plots." It is extremely difficult to understand how the bar plot was built after all.

Response: Thanks for the suggestion. We have re-checked the legend of figures, revised them and tried to make them clearer. For Figure 5, the sentence was modified to “For both (A) and (B), the percentages of common SEGs (intersects of the Venn graphs) in healthy tissues were shown in the bar plots”.

Round 2

Reviewer 2 Report (New Reviewer)

Thank   you  for   your  comments and 

This manuscript is a resubmission of an earlier submission. The following is a list of the peer review reports and author responses from that submission.

Round 1

Reviewer 1 Report

Consolidated Comments on the article "Identification of Human Global, Tissue and within-tissue cell-specific Stably Expressed Genes at Single-Cells Resolution."

The authors identified a set of Stably Expressed Genes (SEGs) with invariant expression in humans across developmental stages, tissues, cell types, and some diseases or stress conditions from single-cell data. The authors produced the microwell-seq-based single-cell RNA-seq datasets, while other datasets were pulled from the NCBI Gene Expression Omnibus database. The authors proposed a pipeline to identify cellular, tissue, and general SEGs from high-throughput single-cell RNA-seq data. The authors identified 408 SEGs from adult and fetal tissues. Then, they compared the SEGs found with their methodology to other omics datasets (from single-cell (10x genomics & smart-seq) and bulky RNA-seq techniques) and found ~ 65% of the identified overlap SEGs between the different methods. Interestingly, the authors identified that the most common SEGs in adult tissue were mainly immune-related genes, while those identified in fetal tissue were spliceosome-related genes. They identified cross-cellular SEGs within 29 human tissues, with thymus and brain tissue showing the most significant SEGs numbers. Additionally, the authors showed the SEGs expressed in healthy tissue also expressed stably in the samples analyzed from tissue under disease or stress conditions. Finally, the authors predicted the composition percentage of major immune cell types across the databases tested by implementing a cell deconvolution model (SEGdecon).

Overall, I consider that the article is well-structured and well-written, and I think the topic discussed will be of interest to the readers of the IJMS journal. However, I noticed some points and definitions/further explanations that need to be addressed before publication.

1.       Section 2.1., paragraph 1, when explaining SEGs identification parameters, refer the readers to the specific section in the methods, e.g., (GESI, Methods, sections 4.3 & 4.4), and 4.5 for the identification across tissues.

2.        Section 2.1., paragraph 2, how does it mean "a set of SEGs remains Spatio-temporal"? It is not clearly defined in the text.

3.       Section 2.1., paragraph 3. The paragraph briefly described the findings in sup. fig2, but it is necessary to give more details to get the home message across. It could be added either in this paragraph or in the figure legend.

4.       Figure 1. (a) make the font size of labels within the figure bigger and improve the resolution of the figure. The labels are unreadable when printed (particularly panels C-E). (b) What does the inset color bar in panel C represents (explain in the figure legend)? (c) what do the axis in C mean? (d) in the figure legend, add the definition of the acronym PCCs.

5.       Sup. Fig.2. (a) explain the gene ratio (how is it calculated?). (b) enhance the description of the figure legend; it needs more details. (c) Explain in the methods "how Fisher exact test with Bonferroni correction was performed."

6.       Section 2.2., paragraph 1, I believe the ratio between Microwell-seq SEGs with Smart-seq and bulky RNA-seq is 266/408 instead of 264/408.

7.       Figure 2. Legend, define/explain what a UMAP means either in the legend or methods.

8.       Sup. Fig 4., and Fig 3C. Would it be possible to add some labeling to the consensus plots shown on the left and top of the heat map?

9.       Section 2.3, Fig. 3C., (a) I would add a discussion/description paragraph to explain the two sets of tissue groups that mostly showed the largest SEGS dataset in adult intestinal tissue. (b) What sets of SEGs are common across all tissue samples ("displayed in red across all the samples in fig. 3C")?

10.   Fig. 4. Legend. Define what a marker SEGs is.

11.   Fig. 5B. add the label for SEGs from disease samples vs. healthy samples as in fig. 5A. Also, add it in the fig legend.

12.   Fig. 6. (a) Increase the font size and the image's resolution; labels are unreadable. (b) move the color code in the pie chart from the top left to the right and make it bigger. (c) Explain why NK/T cells are a combined group in the last two pie charts in the figure legend.

13.   Discussion: Number the groups in the paragraph as:

"traditional TSGs could be defined more clearly with three groups: (1) tissue-specific SEGs with   invariant expression among different cells of the same tissue but with unstable or zero expression for all the other tissues of an organism, (2) tissue SEGs with invariant expression among different cells of the same tissue but not with stable expression for all different tissues of an organism, (3) and tissue-specific genes that are specifically expressed with invariant or varied levels in some of the cells within a tissue."

14.   The links referred in the paper to the project and the SEGdecon do not work!

15.   Discussion, paragraph 3. The figures to be referenced are Fig. 3 A-B and Fig. 3 A., instead of Fig. 2 A-B and Fig. 2 A.

16.   Section 4.2. (a) Define what UMIs mean. (b) it is not clear what m represents.

17.   Section 4.4. Justify the use/selection of the criteria described in this section.

18.   Section 4.8. Add a dash after each letter, as: "Cells were clustered, and T-, B-, NK- and M-cells."

Author Response

The authors identified a set of Stably Expressed Genes (SEGs) with invariant expression in humans across developmental stages, tissues, cell types, and some diseases or stress conditions from single-cell data. The authors produced the microwell-seq-based single-cell RNA-seq datasets, while other datasets were pulled from the NCBI Gene Expression Omnibus database. The authors proposed a pipeline to identify cellular, tissue, and general SEGs from high-throughput single-cell RNA-seq data. The authors identified 408 SEGs from adult and fetal tissues. Then, they compared the SEGs found with their methodology to other omics datasets (from single-cell (10x genomics & smart-seq) and bulky RNA-seq techniques) and found ~ 65% of the identified overlap SEGs between the different methods. Interestingly, the authors identified that the most common SEGs in adult tissue were mainly immune-related genes, while those identified in fetal tissue were spliceosome-related genes. They identified cross-cellular SEGs within 29 human tissues, with thymus and brain tissue showing the most significant SEGs numbers. Additionally, the authors showed the SEGs expressed in healthy tissue also expressed stably in the samples analyzed from tissue under disease or stress conditions. Finally, the authors predicted the composition percentage of major immune cell types across the databases tested by implementing a cell deconvolution model (SEGdecon).

Overall, I consider that the article is well-structured and well-written, and I think the topic discussed will be of interest to the readers of the IJMS journal. However, I noticed some points and definitions/further explanations that need to be addressed before publication.

 Response: Thanks a lot for your comments and questions. We made point-by-point responses to the questions below.

  1. Section 2.1., paragraph 1, when explaining SEGs identification parameters, refer the readers to the specific section in the methods, e.g., (GESI, Methods, sections 4.3 & 4.4), and 4.5 for the identification across tissues.

Response: Thanks for the suggestion. The references have been added in the context.

  1. Section 2.1., paragraph 2, how does it mean "a set of SEGs remains Spatio-temporal"? It is not clearly defined in the text.

Response: Because it was technically difficult to maintain both a high sensitivity and a high stableness (or recurrence) to capture the SEGs among various tissues at different stages with a strict criterion, we adopted a combinatory strategy, i.e., high robustness (and relatively low sensitivity) among the cell types of each sample, but high power (and relatively low consistency) among tissues. An additional condition that should be met with was that the SEGs should be consistent at least in both one adult tissue and one fetal tissue. In each sample, there are many different cell types with different origins. Therefore, by this way, the final SEGs met with the criteria of stable expression in different cells at different developmental stages, and therefore remained spatio-temporal. The ‘non-stableness’ of SEGs in some tissues or samples was not due to their real non-stableness but most occasionally because their stableness was out of detection in these tissues / samples due to large noises. In the previous version of manuscript, the sentences were badly organized, and ‘spatial-temporal’ was changed to ‘spatial’ more accurately: “It should be noted that, the SEGs might be identified from a few rather than all the different tissues, but they remained spatial since (1) each tissue-SEG was required strictly to be stably expressed among all cell types, and among cell clusters for each cell type, at least in one sample from the corresponding tissue, and (2) there were multi-origin cell types within each sample.”

  1. Section 2.1., paragraph 3. The paragraph briefly described the findings in sup. fig2, but it is necessary to give more details to get the home message across. It could be added either in this paragraph or in the figure legend.

Response: More details and comments have been added in the Paragraph 3 of Section 2.1 so that the main point can be delivered better now: “The results are consistent with previous observations and functional expectation on HKGs, suggesting that SEGs and HKGs could show apparent overlap in both the composition of genes and the essential function [1-2].”

  1. Figure 1. (a) make the font size of labels within the figure bigger and improve the resolution of the figure. The labels are unreadable when printed (particularly panels C-E). (b) What does the inset color bar in panel C represents (explain in the figure legend)? (c) what do the axis in C mean? (d) in the figure legend, add the definition of the acronym PCCs.

Response: Thanks for the comments, and we have made corresponding revisions on the figure and legend. The figure inserted in the manuscript was not original and therefore the resolution was limited, and the problem will be resolved once the original figures are used. (a) The font size of labels has been changed. (b) - (c) For panel C, more explanation and an literature about UMAP were added in the legend: “The single cells were clustered according to the gene expression profile, and shown as points in a 2-dimension plane of Uniform Manifold Approximation and Projection (UMAP) [35]. The gene expression level was represented by the color shown in the scale bar”. (c) The axis’s represented two UMAP dimensions, which were like PCA and explained in detail in literature 35. Like t-sne, It has been now widely used in single-cell RNA-seq data representation. (d) The definition of PCCs has been added.

  1. Sup. Fig.2. (a) explain the gene ratio (how is it calculated?). (b) enhance the description of the figure legend; it needs more details. (c) Explain in the methods "how Fisher exact testwith Bonferroni correction was performed."

Response: The analysis was performed with a widely applied R package ‘clusterProfiler’. Gene Ratio was calculated as the ratio of SEGs with corresponding GO function among the total SEGs. Details were added in the legend as “GeneRatio represented the ratio of SEGs with corresponding GO function among the total SEGs. Fisher exact tests with Bonferroni correction were performed for the functional enrichment analysis. R package ‘clusterProfiler’ was used for the functional enrichment analysis and dot plot preparation (http://bioconductor.org/packages/release/bioc/html/clusterProfiler.html)”.

  1. Section 2.2., paragraph 1, I believe the ratio between Microwell-seq SEGs with Smart-seq and bulky RNA-seq is 266/408 instead of 264/408.

Response: Thanks for pointing out the error. It was corrected.

  1. Figure 2. Legend, define/explain what a UMAP means either in the legend or methods.

Response: Thanks for the suggestion. We have explained UMAP in the legend of Figure 1. Please refer to the response to the Question 4.

  1. Sup. Fig 4., and Fig 3C. Would it be possible to add some labeling to the consensus plots shown on the left and top of the heat map?

Response: Thanks for the suggestion. Labeling was made to the figures, and the legend was revised accordingly.

  1. Section 2.3, Fig. 3C., (a) I would add a discussion/description paragraph to explain the two sets of tissue groups that mostly showed the largest SEGS dataset in adult intestinal tissue. (b) What sets of SEGs are common across all tissue samples ("displayed in red across all the samples in fig. 3C")?

Response: Thank you very much for the suggestion. We added a Supplemental Table (Dataset S1, Sheet 4) and a paragraph in the corresponding section of RESULTS to describe the common set of SEGs, heterogeneity of SEGs in human heart cells or Erythroid cells, and the two subsets of adult intestine/stomach cells. Please refer to the added paragraph 3 in Section 2.3.

  1. Fig. 4. Legend. Define what a marker SEGs is.

Response: Definition was added accordingly. “Marker SEGs represent SEGs that are expressed specifically or with striking higher level in indicated cell types and can be used as gene markers for the corresponding cells.”

  1. Fig. 5B. add the label for SEGs from disease samples vs. healthy samples as in fig. 5A. Also, add it in the fig legend.

Response: Thanks for the suggestion. Labels were added in Fig 5 accordingly.

  1. Fig. 6. (a) Increase the font size and the image's resolution; labels are unreadable. (b) move the color code in the pie chart from the top left to the right and make it bigger. (c) Explain why NK/T cells are a combined group in the last two pie charts in the figure legend.

Response: Thanks for the suggestions. The font size was increased. The color code was moved to the right. The resolution has been increased for the image. NK and T cells in peripheral blood mononuclear cells (PBMCs) often show similar gene expression profile, and in some samples, the two types of cells cannot accurately distinguished. In the original report for the PBMCs of tuberculosis cases, the NK and T cells were considered as a big combined group (Cai, Y., et al., EBioMedicine, 2020. 53: p. 102686). That is why we pooled them together. Explanation and the reference were added in the RESULTS section 2.6: “It should be noted that the NK/T cells are difficult to separate from the scRNA-seq data of TB and LTBI samples, and therefore merged into a unique cell type [34]”.

  1. Discussion: Number the groups in the paragraph as:

"traditional TSGs could be defined more clearly with three groups: (1) tissue-specific SEGs with   invariant expression among different cells of the same tissue but with unstable or zero expression for all the other tissues of an organism, (2) tissue SEGs with invariant expression among different cells of the same tissue but not with stable expression for all different tissues of an organism, (3) and tissue-specific genes that are specifically expressed with invariant or varied levels in some of the cells within a tissue."

Response: Numbers were added accordingly.

  1. The links referred in the paper to the project and the SEGdecon do not work!

Response: The links were corrected and now they work. http://61.160.194.165:3080/hSEGdb/index.html.

  1. Discussion, paragraph 3. The figures to be referenced are Fig. 3 A-B and Fig. 3 A., instead of Fig. 2 A-B and Fig. 2 A.

Response: Corrected.

  1. Section 4.2. (a) Define what UMIs mean. (b) it is not clear what m represents.

Response: The definition of UMIs (Unique Molecular Identifiers) was added.

  1. Section 4.4. Justify the use/selection of the criteria described in this section.

Response: Before the criteria were selected, we tested the different combination of parameters for three important factors influencing the sensitivity and accuracy of the final inter-cluster SEGs. We found that the strict within-cluster SEGs influenced the sensitivity intensively while for most known HKGs in most samples, the ratio of / between clusters was smaller than 0.5. Therefore, we adopted a strategy of the combined parameters with relaxed criterion for within-cluster SEGs, strict criterion for the recurrence of within-cluster SEGs among different clusters, and a moderately strict cutoff / to ensure the expression stableness of the SEGs among clusters. We have added detailed explanation in the method.

  1. Section 4.8. Add a dash after each letter, as: "Cells were clustered, and T-, B-, NK- and M-cells."

Response: Revised accordingly.

Reviewer 2 Report

The article would investigate the expression stability of different genes in single cells from different cell/organs also in the presence of diseases. The research question appears too vague and it is hard to understand for this reviewer the clinical relevance of such investigation. I believe the article has to be rejected and I would recommend authors to better focus of clinical relevance for researches.

Author Response

Thanks for your comments.

Reviewer 3 Report

This paper describes nicely the identification of SEG using multiple published datasets.

This paper is very interesting, generally easy to follow, and results are convincing.

One major comment is that scRNAseq or bulk RNAseq analysis usually doesn't use "housekeeping" genes for normalization but rather total UMI or rpkm so the only direct application of SEG I can think of would be more having qPCR housekeeping genes for each tissue or multiple tissue.

Hence the manuscript would benefit more on comparing previously used/published housekeeping genes and confirm or not their quality of SEG.

Also, I think the authors have now a perfect knowledge on SEG and should test some of the predicted SEGs by qPCR and add a selected confirmed list of the best SEGs for each tissue tested and general SEGs across tissue (table S1 and S2 are good but so many to choose from, are some better than other etc.)

This confirmation by qPCR of a subset would really benefit the paper and especially its citation rate as they would provide a clear validated list of new housekeeping genes, that can be used in many studies.

Finally, even if the majority of the manuscript is well written, I could not understand some parts, I asked other colleagues, but none of them could clearly understand as well.  So please fully re-write the cited sentences below:

"It was interesting to find that the most reliable adult-stage specific SEGs, which were common in no fewer than 5 fetal tissues but not present in any adult tissue, were mainly immune-related genes"

? Confusing

Here nothing make sense to me:

"For each individual tissue, followed by analysis of the within-tissue cell-type specific

SEGs. Specific SEGs could be identified for most cell types, and yet only a few of them

showed strikingly higher expression level than other cell types so as to be capable of serving

as markers for the corresponding cells within the tissue (Fig. 4B-C; Supplemental Dataset

S2)." ?

Fig2 D don't understand the legend what is what ?

In the M&M

4.2. Binning of single cells

why using letters with you say the letters are actually 10, 100 and 100? It is confusing

4.8. Cell decomposition models

define the coefficient parameter

"represented the faction of cell type c in the testing dataset "? what is faction

define "Es was the total expression of the sth cell-specific SEG in the testing dataset"

What is total expression ? explain how you calculate it from scRNAseq dataset

link http://61.160.194.165/hSEGdb not working

Author Response

This paper describes nicely the identification of SEG using multiple published datasets.

This paper is very interesting, generally easy to follow, and results are convincing.

One major comment is that scRNAseq or bulk RNAseq analysis usually doesn't use "housekeeping" genes for normalization but rather total UMI or rpkm so the only direct application of SEG I can think of would be more having qPCR housekeeping genes for each tissue or multiple tissue. Hence the manuscript would benefit more on comparing previously used/published housekeeping genes and confirm or not their quality of SEG.

Response: Thanks a lot for the suggestion, which is very helpful. We annotated the frequently used HKGs or commercial HKGs as qRT-PCR reference genes, and the specific endogenous reference genes for specific tissues, and compared them with the SEGs identified in this research. We did find that many of them were not among the top list of stably expressed genes. We added a paragraph in DISCUSSION in the revised manuscript to clarify this:

“To serve as inner control genes for quantitative RT-PCR (qRT-PCR) could be an-other important application of SEGs. However, the commercial set of human common HKGs, which are also frequently used as qRT-PCR inner controls in laboratory experi-ments, are often not among the SEG list we identified at single-cell resolution. For ex-ample, six among the fifteen endogenous reference genes of Applied Biosystems (http://www.appliedbiosystems.com) were not covered by the global SEGs or tis-sue-SEGs, including GUSB, HPRT1, TBP, IPO8, POLR2A and HMBS. Genes identified as reference genes for qRT-PCR normalization in specific tissues are also frequently recog-nized as not stably expressed in this research. For instances, SDHA and TBP in human bladder cancers [36], CYC1 and TOP1 in human brains [37], and TRAP1, FPGS and DECR1 in human PBMC samples [38], were identified as tissue HKGs that could be used as endogenous control genes, but none of them is with stable expression in corresponding tissues or cells analyzed in this study. The genes listed here are not necessarily non-SEGs, but did show larger variance than other SEGs in the included samples. In practice, 32 spatial-temporal SEGs that are repeatedly identified from all the cell types should be more reliable and used as the first choices for qRT-PCR experiments (Supple-mental Dataset S1, Cell_type_SEGs, the top 32 genes). More emphasis should be placed in validation of more reliable SEGs in different normal and diseased tissues.”

Also, I think the authors have now a perfect knowledge on SEG and should test some of the predicted SEGs by qPCR and add a selected confirmed list of the best SEGs for each tissue tested and general SEGs across tissue (table S1 and S2 are good but so many to choose from, are some better than other etc.) This confirmation by qPCR of a subset would really benefit the paper and especially its citation rate as they would provide a clear validated list of new housekeeping genes, that can be used in many studies.

Response: Many thanks for the great suggestion. Verification of the SEGs using qPCR would be very useful, and actually we happened to have proposed a continuing project to annotate a list of high-quality and highly reliable tissue-SEGs with a combined strategy of computational prediction and experimental validation. However, it is really difficult for us to collect multiple human samples to perform the experiments during the allowed short period of revision. By observing the distribution of cell type SEGs (newly added sheet 4 in Supplemental Dataset S1), we found that 32 were always SEGs for all of the cell types analyzed in this study, and all these 32 genes were among the spatial-temporal SEG list (in yellow background). These SEGs could be more reliable ones that could be served as the first choices for qRT-PCR inner controls. We also added comments in DISCUSSION. Please refer to the response to the last question.

Finally, even if the majority of the manuscript is well written, I could not understand some parts, I asked other colleagues, but none of them could clearly understand as well.  So please fully re-write the cited sentences below:

Response: Thank for the comments. We have revised the manuscript substantially.

"It was interesting to find that the most reliable adult-stage specific SEGs, which were common in no fewer than 5 fetal tissues but not present in any adult tissue, were mainly immune-related genes"

? Confusing

Response: Thank you so much for pointing out this mistake. It was corrected. “It was interesting to find that the most reliable adult-stage specific SEGs, which were common in no fewer than 5 adult tissues but not present in any fetal tissue, were mainly immune-related genes”.

Here nothing make sense to me:

"For each individual tissue, followed by analysis of the within-tissue cell-type specific

SEGs. Specific SEGs could be identified for most cell types, and yet only a few of them

showed strikingly higher expression level than other cell types so as to be capable of serving

as markers for the corresponding cells within the tissue (Fig. 4B-C; Supplemental Dataset

S2)." ?

Response: There was a typo which could influence the meaning of the sentences, and now it was corrected: “…, followed by analysis of the within-tissue cell-type specific SEGs, specific SEGs could be identified for most cell types, and yet only a few of them showed strikingly higher expression level than other cell types so as to be capable of serving as markers for the corresponding cells within the tissue…”. For most cell types in each tissue, a list of specific SEGs could be identified. However, these SEGs could also be expressed in other cell types but not stably. To serve as marker genes of a certain cell type, a SEG must be expressed specifically or with significantly higher levels in the corresponding cell type in the first place. We also defined ‘Marker SEGs’ more clearly in the legend of Fig. 4 in the revised manuscript.

Fig2 D don't understand the legend what is what ?

Response: Thanks a lot for pointing out the problem. We have revised the legend: “Expression of two representative SEGs/HKGs specifically detected by Microwell-Seq (RPL7 and RPS29), Smart-Seq (CHD1 and SRPK1) or bulky RNA-seq (TUBD1 and ERCC5) in single-cell clusters of a representative sample. The representative sample (an adult lung sample) was selected from [30] randomly”.

In the M&M

4.2. Binning of single cells

why using letters with you say the letters are actually 10, 100 and 100? It is confusing

Response: The numbers were set after testing and optimization. We have added description on this in the revised manuscript: “In this research, n, m and r were optimized according to the number of cell clusters that can be used for further analysis, the overall stableness of gene expression among bins and the power of SEGs identified from the cell clusters, and eventually set as 10, 100 and 100 respectively”.

4.8. Cell decomposition models

define the coefficient parameter

Response: We have revised the sentences: “where c represented a immune cell type from the space of [T, B, NK and M], wsc was the coefficient parameter and represented by the average expression level of the sth cell-specific SEG in cell type c of the training dataset, fc represented the fraction of cell type c in the testing dataset, and Es was the general expression of the sth cell-specific SEG in the testing dataset calculated as the average UMIs among all the analyzed single cells”.

"represented the faction of cell type c in the testing dataset "? what is faction

Response: Sorry for the typo. We have revised ‘faction’ to ‘fraction’.

define "Es was the total expression of the sth cell-specific SEG in the testing dataset"

What is total expression ? explain how you calculate it from scRNAseq dataset

Response: We have revised the sentences: “where c represented a immune cell type from the space of [T, B, NK and M], wsc was the coefficient parameter and represented by the average expression level of the sth cell-specific SEG in cell type c of the training dataset, fc represented the fraction of cell type c in the testing dataset, and Es was the general expression of the sth cell-specific SEG in the testing dataset calculated as the average UMIs among all the analyzed single cells”.

link http://61.160.194.165/hSEGdb not working

Response: Thanks for pointing out the error. The link has been corrected and now it works. http://61.160.194.165:3080/hSEGdb.

Round 2

Reviewer 2 Report

No improvements.

Reviewer 3 Report

Thank you for your careful edits.

I understand that qPCR validation on many human sample can be quite a hassle to obtain within a limited period of time.

Good luck in your future projects  

Regards,